# *Mycobacterium tuberculosis* Rv2005c Induces Dendritic Cell Maturation and Th1 Responses and Exhibits Immunotherapeutic Activity by Fusion with the Rv2882c Protein

**DOI:** 10.3390/vaccines8030370

**Published:** 2020-07-10

**Authors:** Yong Woo Back, Ki Won Shin, Seunga Choi, Hye-Soo Park, Kang-In Lee, Han-Gyu Choi, Hwa-Jung Kim

**Affiliations:** Department of Microbiology, of Medical Science, College of Medicine, Chungnam National University, Daejeon 35015, Korea; lenpk@nate.com (Y.W.B.); rldnjs7572@naver.com (K.W.S.); seungachoi@cnu.ac.kr (S.C.); 01027192188@hanmail.net (H.-S.P.); popigletoh@nate.com (K.-I.L.)

**Keywords:** *Mycobacterium tuberculosis*, hypoxia, DosR, chemo-immunotherapy, multifunctional T cell

## Abstract

Immunotherapy represents a promising approach for improving current antibiotic treatments through the engagement of the host’s immune system. Latency-associated antigens have been included as components of multistage subunit tuberculosis vaccines. We first identified Rv2005c, a DosR regulon-encoded protein, as a seroreactive protein. In this study, we found that Rv2005c induced dendritic cell (DC) maturation and Th1 responses, and its expression by *Mycobacterium tuberculosis* (Mtb) within macrophages was enhanced by treatment with CoCl_2_, a hypoxia-mimetic agent. T cells activated by Rv2005c-matured DCs induced antimycobacterial activity in macrophages under hypoxic conditions but not under normoxic conditions. However, Rv2005c alone did not exhibit any significant vaccine efficacy in our mouse model. The fusion of Rv2005c to the macrophage-activating protein Rv2882c resulted in significant activation of DCs and antimycobacterial activity in macrophages, which were enhanced under hypoxic conditions. Furthermore, the Rv2882c-Rv2005c fusion protein showed significant adjunctive immunotherapeutic effects and led to the generation of long-lasting, antigen-specific, multifunctional CD4^+^ T cells that coproduced TNF-α, IFN-γ and IL-2 in the lungs of our established mouse model. Overall, these results provide a novel fusion protein with immunotherapeutic potential as adjunctive chemotherapy for tuberculosis.

## 1. Introduction

Tuberculosis (TB), an infectious disease caused by *Mycobacterium tuberculosis* (Mtb), remains a major global health problem that urgently needs to be controlled because 1.67 million people died of TB in 2016 [1]. The appearance of drug-resistant TB and the imperfect protection provided by the *M. bovis* BCG (Bacillus Calmette–Guérin) vaccine, which is currently the only TB vaccine, have led to the development of a more effective TB vaccine [2]. In particular, vaccines for adjunctive immunotherapy are urgently needed to shorten the duration of chemotherapy and to prevent the reactivation of latent TB.

There are still several challenging issues in TB treatment, including long treatment periods, drug resistance, and various side effects [3,4]. In the case of TB, increasing numbers of studies have been conducted to develop immunotherapeutics, particularly with regard to the fight against MDR-TB [5]. Strategies aimed at harnessing a patient’s immune response to Mtb may protect actively infected individuals against disease progression and might accelerate the cure and/or increase the cure rate achieved by antibiotic therapy [5]. Although the development of more effective prophylactic vaccines for TB is a high priority, therapeutic approaches, such as postexposure vaccines, which could be used in combination with antibiotics to shorten treatment regimens, clear bacilli [6], and limit the spread of MDR-TB [7,8,9], should be explored. In this regard, the therapeutic efficacy of a few tuberculosis vaccine candidates has been evaluated [10,11,12,13].

Dendritic cells (DCs) are very specialized antigen-presenting cells of the immune system; these cells play a crucial role in antigen presentation and act as a link between innate and adaptive immunity [14]. When Mtb is inhaled into the lung, DCs phagocytose bacteria and antigens [15]. After antigen uptake, DCs migrate to secondary lymphoid tissues, and DCs become differentiated [16]. DCs that express MHC molecules and costimulatory molecules undergo maturation and are able to activate T cells [17], which form granulomas with infected monocytes and macrophages to restrict Mtb growth [18]. The environment inside granulomas is hypoxic and nutrient deficient [19,20,21]. It is possible that the ideal vaccine targets are antigens that are preferentially expressed by Mtb after adaptation to and long-term persistence in environments such as granulomas. T cells activated by DCs stimulate macrophages to kill intracellular bacteria.

Although many mycobacterial proteins have been reported to activate the immune system, little is known about the protective role of these proteins in host defense against TB [22]. Several novel TB vaccine candidates have been evaluated in clinical trials [23]. TB vaccine research has been focused on the development of a multistage subunit vaccine consisting of T cell antigens that are secreted by logarithmically growing Mtb and antigens that are related to dormancy [24]. We identified mycobacterial antigens with strong immunoreactivity from multidimensional fractions of Mtb culture filtrate [22]. We found that Rv2005c reacted with the sera of active TB patients. Rv2005c is a member of the family of universal stress proteins [25], a DosR regulon-encoded protein, a latency-associated antigen [26,27,28] and a predicted vaccine candidate [29]. In this study, we investigated the effect of Rv2005c on DCs and its vaccine potential. Rv2005c induced DC maturation and Th1 polarization, which elicited antimycobacterial activity in macrophages under hypoxic conditions; however, Rv2005c did not show any significant vaccine efficacy in a mouse model. However, the fusion of Rv2005c to Rv2882c, which is a macrophage-activating protein [22], exhibited significant therapeutic potential as adjunctive chemotherapy in a mouse model of TB infection.

## 2. Material and Methods

### 2.1. Mice

Specific pathogen-free (SPF) female WT C57BL/6 mice, TLR2 knockout (KO) mice, TLR4 KO mice, and OT-2 TCR transgenic mice (all at 5–6 weeks of age) were purchased from the Jackson Laboratory (Bar Harbor, ME, USA). All animals were maintained under SPF barrier conditions at the Medical Research Center, Chungnam National University (Daejeon, Korea). All animals were kept under controlled conditions, sterilized food and water was provided ad libitum. All the animal experiments were approved by the Institutional Research and Ethics Committee at Chungnam National University (approval number: 201903A-CNU-5). All the animal procedures were performed in accordance with the guidelines of the Korean Food and Drug Administration.

### 2.2. Infection, Bacterial Strains and Cell Preparations

Mtb H37Rv, Mtb H37Ra and *M. bovis* BCG Tokyo were grown in Middlebrook 7H9 (Difco Laboratories, Detroit, MI) supplemented with 0.5% glycerol, 0.05% Tween-80 (Sigma, St. Louis, MO, USA), and 10% OADC enrichment (Sigma).

Bone marrow-derived macrophages (BMDMs) were differentiated as described [30]. Briefly, flushed from the femurs of mice, then RBC were lysed. After washing the cells, the total cells were suspended in Dulbecco’s modified Eagle’s medium (DMEM) (Welgene Co., Daegu, Korea) containing 10% fetal bovine serum, 50 ng/mL mouse macrophage colony-stimulating factor (M-CSF; Rocky Hill, NJ, USA), and 1% antibiotics (Welgene). Finally, cells were plated in 100-mm plates and incubated for 7 days at 37 °C in 5% CO_2_. 

Bone marrow-derived dendritic cells (BMDCs) were cultured at 37 °C in the presence of 5% CO_2_ using Roswell Park Memorial Institute (RPMI) 1640 media supplemented with 10% FBS, 1% antibiotics (Welgene), 0.1% 2-mercaptoethanol, 5 mM HEPES buffer, 1% MEM solution, 20 ng/mL granulocyte-macrophage colony-stimulating factor (GM-CSF) and 2 ng/mL IL-4. The nonadherent cells and loosely adherent proliferating DC aggregates were harvested on day 7 or 8 and were used for further experiments.

Lungs and spleen were isolated as described [30]. Briefly, the lung cut into several pieces, and agitated in 5 mL cell dissociation buffer for 15 min at 37 °C. Then, the lung and spleen were filtered through a 100-μm cell strainer in RPMI medium (Welgene). After RBC were lysed, and CD4^+^ T cells were isolated from the lung and spleen of mice using a MACS column. 

### 2.3. Purification of Recombinant Protein

To produce the recombinant Rv2882c-Rv2005c protein and the individual proteins, the corresponding genes were amplified by PCR using Mtb H37Rv ATCC27294 genomic DNA as the template and the following primers: Rv2882c-Rv2005c, Rv2882c forward, 5′-GAATTCGATGATTGA- TGAGGCTCTCTTC-3′; Rv2882c reverse, 5′-AAGCTTGACCTCCAGCAGCTCGCCTTCC-3′; Rv2005c forward, 5′-AAGCTTATGTCTAAACCCCGCAAGCAG-3′; Rv2005c reverse, 5′-GCGGCC- GCCGACTGCCGTGCCACGATCAC-3′; Rv2882c forward, 5′-CATATGATTGATGAGGCTCTCTT- CGAC-3′; Rv2882c reverse, 5′-AAGCTTGACCTCCAGCAGCTCGCCTTC-3′; Rv2005c forward, 5′-CATATGTCTAAACCCCGCAAGCAGCAC-3′; and Rv2005c reverse, 5′-AAGCTTCGACTGCCGT- GCCACGATCAC-3′. The Rv2882c-Rv2005c PCR product was digested with *EcoR I*, *Hind III* and *Not I*, Rv2882c was digested with *Nde I* and *Hind III*, and Rv2005c was digested with *Nde I* and *Hind III*. The products were inserted into the pET22b (+) vector (Novagen, Madison, WI, USA), and the resultant plasmid was sequenced. The result was Rv2882c-Rv2005c. Every recombinant plasmid was transformed into *E. coli* BL21 cells.

Recombinant Rv2882c-Rv2005c was purified as described previously [30]. Briefly, *E. coli* cells were grown in a 37 °C shaking incubator. Then, *E.coli* were grown to an OD of ~0.4 at 600 nm and were incubated with 0.1 mM isopropyl-D-thiogalactopyranoside (IPTG; ELPIS-Biotech, Daejeon, Korea). After 4 to 6 h, the bacterial cells were harvested and sonicated on ice. The *E. coli* was centrifuged and the supernatant containing the protein was then purified. Each purification fraction was analyzed by sodium dodecyl sulfate polyacrylamide gel electrophoresis (SDS-PAGE) followed by Coomassie brilliant blue (CB) staining and western blot (WB) analysis using anti-His antibodies (Santa Cruz). The protein concentration was estimated using a bicinchoninic acid protein assay kit (Pierce, Rockford, IL, USA) with bovine serum albumin (BSA) as the standard. The purity of every protein was evaluated by CB staining and WB using an anti-histidine antibody.

### 2.4. Cytotoxicity Analysis

Rv2005c (10 μg/mL), CoCl_2_ (300 μM), or Rv2005c (10 μg/mL) with CoCl_2_ (300 μM) was added to isolated BMDMs cultured in 12-well plates (5 × 10^5^ cells/mL) to investigate the cytotoxic effects. After 24 h of treatment, the harvested BMDMs were washed with PBS, stained with FITC-Annexin V and PI (BD Biosciences), and then analyzed using a FACS Canto flow cytometer (BD Biosciences).

### 2.5. Enzyme-Linked Immunosorbent Assay (ELISA)

A sandwich enzyme-linked immunosorbent assay was used to detect the TNF-α, IL-1β, IFN-γ, IL-2, IL-4 and IL-12p70 (eBioscience) concentrations in the culture media as described previously [22]. 

### 2.6. Flow Cytometry Analysis

BMDCs were stimulated with Rv2005c (10 μg/mL) for 24 h. The stimulated cells were harvested and washed with PBS. The BMDCs were stained with PE-conjugated anti-CD40, anti-CD80, anti-CD86, anti-MHC class I and anti-MHC class II and with FITC-conjugated CD11c (eBioscience) for 30 min at 4 °C. The cells were washed with PBS and suspended in 250 μL PBS. The fluorescence was measured by flow cytometry, and the data were processed using FlowJo software.

### 2.7. Western Blot Analysis

After stimulation with Rv2005c (10 μg/mL) over time, the culture media was removed, and the BMDCs were incubated in lysis buffer (50 mM Tris-HCl pH 7.5, 150 mM NaCl, 1% Triton X-100, 2 mM EDTA, 0.1% SDS, 1% sodium deoxycholate, and protease inhibitor cocktail) for 30 min on ice. The concentration of the cellular proteins was determined by the Bradford assay, and equal amounts of proteins were separated by SDS-PAGE. The separated proteins were transferred to polyvinylidine difluoride (PVDF) membranes. The membranes were blocked with 5% skim milk and incubated with primary antibodies overnight. The membranes were washed 3 times for 10 min with PBS containing 0.1% Tween 20 (PBS/T) and were incubated with HRP-conjugated secondary antibodies for 1 h at room temperature. An enhanced chemiluminescence system (ECL; Millipore Corporation, Billerica, MA, USA) was used, followed by exposure to chemiluminescence film to visualize the proteins.

### 2.8. In Vitro T Cell Proliferation Assay

Responder T cells, which participate in naïve T cell reactions, were isolated from the total mononuclear cells extracted from BALB/c mice using a MACS column (Miltenyi Biotec). OVA-specific responder CD4^+^ T cells were obtained from the splenocytes of OT-2 mice and these T cells were stained with 1 μM CFSE (Invitrogen). Then, T cell proliferation assay was performed as described previously [31]. Briefly, DCs (2 × 10^5^ cells per well) were treated with the OVA peptide and 10 μg/mL Rv2005c for 24 h. Then, cocultured with CFSE-stained T cells at a DC at a DC:T cell ratio of 1:10. After 3 or 4 days of coculture, each T cell batch was stained with PerCP-Cy5.5-conjugated anti-CD4^+^ mAb and analyzed by flow cytometry. The supernatants were harvested measured by ELISA (eBioscience).

### 2.9. Hypoxia Induced in Bmdms by Cocl_2_

#### 2.9.1. Cytotoxicity Analysis

Cobalt chloride (CoCl_2_) (300 μM) was added to isolated BMDMs cultured in 12-well plates (5 × 10^5^ cells/mL) to investigate the cytotoxic effect of CoCl_2_. After 24 h of treatment, the harvested BMDMs were washed using PBS, stained with FITC-Annexin V and PI (BD Biosciences), and then analyzed using a FACS Canto flow cytometer (BD Biosciences).

#### 2.9.2. Western Blot Analysis

After 24 h of treatment of BMDMs with CoCl_2_ (Sigma), the culture media was removed, and the BMDMs were incubated in lysis buffer (50 mM Tris-HCl pH 7.5, 150 mM NaCl, 1% Triton X-100, 2 mM EDTA, 0.1% SDS, 1% sodium deoxycholate, and protease inhibitor cocktail) for 30 min on ice. The concentration of the cellular proteins was determined by the Bradford assay, and equal amounts of proteins were separated by SDS-PAGE. The separated proteins were transferred to PVDF membranes. The membranes were blocked with 5% skim milk and incubated with primary antibodies overnight at cold temperatures. The membranes were washed 3 times for 10 min with PBS containing 0.1% Tween 20 (PBS/T) and incubated with the HRP-conjugated secondary antibodies for 1 h at room temperature. An enhanced chemiluminescence system (ECL; Millipore Corporation, Billerica, MA, USA) was used, followed by exposure to chemiluminescence film to visualize the proteins.

#### 2.9.3. Hypoxia-induced Intracellular Survival of Mtb

BMDMs were infected with Mtb at a multiplicity of infection of 1:1 (bacteria to BMDMs) for 4 h. The infected BMDMs were treated with amikacin (200 μg/mL) for 2 h and then washed twice with PBS. BMDCs were treated with Rv2005c (10 μg/mL), Rv2882c-Rv2005c (0.5 μg/mL) or LPS (100 ng/mL) for 24 h and then coincubated with splenocytes for 3 days. The splenocytes were activated with PMA (100 nM) and Ionomycin (1000 nM) for 2 h before coculture, and these cells were considered to be pharmacologically activated T cells. The DC-splenocyte populations were cocultured with the infected BMDMs for 3 days. The macrophage cultures were immediately dissolved in 0.1% saponin to determine the bacterial burden. Then, the supernatants and cell lysates were serially diluted and plated onto Middlebrook 7H10 agar supplemented with 10% OADC, 10 μg/mL amphotericin B (Sigma). After 2–3 weeks, the colony-forming units (CFUs) were counted to determine the number of colonies on plates. 

### 2.10. Vaccine Evaluation

#### 2.10.1. In Vivo Chemotherapy Assay

To induce progressive pulmonary TB, mice (*n* = 15/group) were anesthetized with 1.2% 2,2,2-tribromoethanol (avertin), exposed via a small midline incision, and inoculated intratracheally (I.T.) with 1 × 10^4^ CFUs of Mtb H37Ra in 50 μL saline [32]. To evaluate the protective efficacy of the Rv2882c-Rv2005c fusion protein and the immune responses generated before and after the course of immunotherapy with this antigen and chemotherapy in Mtb-infected animals, mice were randomized (5 animals per group) and treated with Rv2882c-Rv2005c, adjuvant alone or the combination of Rv2882c-Rv2005c and adjuvant. The mice receiving immunotherapy with Rv2882c-Rv2005c were administered 5 μg/dose/mouse protein emulsified in Dimethyldioctadecylammonium bromide (DDA; 250 μg/dose/mouse) and Monophosphoryl Lipid A (MPL; 25 μg/dose/mouse). The control groups were immunized with adjuvant alone. In the ’short duration’ chemotherapy experiment, after 3 weeks of infection, the mice were given isoniazid (INH, 0.1 g/L) and rifampin (RIF, 0.1 g/L) in their drinking water ad libitum for 4 weeks [33]. These mice were sacrificed 3, 6, 9, 12 or 20 weeks later, and the lungs were isolated and used for CFU, ELISA, and FACS analyses. 

#### 2.10.2. Analysis of Antigen-Specific T Cell Cytokine Secretion

The mice were sacrificed at 6, 9, 12 or 20 weeks after infection to analyze the IFN-γ, IL-2, IL-4, and IL-17 levels in the mouse lungs by ELISA. The cells from each lung (5 × 10^5^ cells/well) were incubated in triplicate in individual wells of 48-well microtiter plates for 72 h at 37 ℃ with either RPMI 1640+10% FBS medium (negative control) or with Rv2882c-Rv2005c (5 μg/mL). Analyses of the cytokines in the culture medium were performed as recommended by the manufacturer (eBioscience). The levels of the cytokines released into the culture medium were determined by measuring the absorbance at 450 nm with a microplate reader.

#### 2.10.3. Intracellular Cytokine Assays

For intracellular cytokine staining, single-cell suspensions from the lung cells (2 × 10^6^ cells) were stimulated with each antigen (5 μg/mL) for 12 h at 37 °C in the presence of GolgiStop (BD Biosciences). Then, cells were harvested and washed. The cells were first blocked with Fc Block (anti-CD16/32) and then stained with BV605-conjugated anti-CD4, FITC-conjugated anti-CD62L, and PE-conjugated anti-CD44 Abs, fixed and permeabilized using a Cytofix/Cytoperm kit (BD Biosciences), and stained with PE Cy7-conjugated anti-TNF-α, APC-conjugated anti-IL-2, and PE-conjugated anti-IFN-γ Abs as described previously [30]. Intracellular cytokine levels were detected on the software program Novocyte (Acea Biosiences, San Diego, CA, USA) and analyzed using the software program FlowJo.

#### 2.10.4. Bacterial Count Analysis

Mice were euthanized at 3, 6, 9, 12 or 20 weeks by CO_2_ asphyxiation, after H37Ra challenge. Then, the lungs of the individual mice were homogenized and serially diluted in PBS. The number of viable bacteria was determined on 7H10 agar supplemented with 10% OADC, 10 μg/mL amphotericin B (Sigma) and 2 μg/mL 2-thiophenecarboxylic acid hydrazide (Sigma). The bacterial numbers were counted after 2–3 weeks of incubation at 37 ℃. The protective efficacies are expressed as the log10 reduction in the bacterial count in immunized mice compared with the bacterial counts in each group.

### 2.11. Statistical Analysis

All the experiments were performed at least three times and results were analyzed using GraphPad Prism 4.03 (GraphPad Software, San Diego, CA, USA). The significance of the differences between samples were evaluated using one-way ANOVA followed by Tukey’s multiple comparison analysis. The data in the graphs are expressed as the mean ± SEM. Differences with values of * *p* < 0.05, ** *p* < 0.01, or *** *p* < 0.001 were considered statistically significant. 

## 3. Results

### 3.1. The Recombinant Rv2005c Protein Induces the Maturation and Activation of Dcs 

Proteins from Mtb culture filtrates were fractionated by biochemical chromatography as previously described, and the immunoreactivity of each fraction was measured [22]. The protein that strongly reacted with the sera of active TB patients was identified as Rv2005c by liquid chromatography-electrospray ionization mass spectrometry (LC-ESI/MS) as previously described [34] (Appendix A). To define the immunological effect of Rv2005c on DCs, first, the protein was expressed in *E. coli* BL21 and purified by His-affinity chromatography. The purified Rv2005c protein showed a major single band at 30 kDa by SDS-PAGE and strongly reacted with anti-His antibody in immunoblotting (Figure 1A). Endotoxin content of prepared Rv2005c was below 15 pg/mL (<0.1 UE/mL) according to an LAL assay. Next, we investigated whether Rv2005c could induce DC maturation. BMDCs treated with Rv2005c significantly increased the expression of the costimulatory molecules CD40, CD80 and CD86, as well as MHC class II molecules, in a dose-dependent manner (Figure 1B). Rv2005c-stimulated BMDCs also secreted high levels of IL-1β, TNF-α, and IL-12p70 in a dose-dependent manner, whereas untreated BMDCs secreted negligible amounts of these cytokines (Figure 1C). The activity of Rv2005c at a concentration of 5 µg/mL was comparable to that of LPS (100 ng/mL), which was used as a positive control. It is well known that MAPKs and NF-κB are important signaling molecules for the regulation of DC maturation and for the secretion of proinflammatory cytokines [30]. We therefore examined the activation of MAPKs and NF-κB in response to Rv2005c. As expected, Rv2005c triggered the phosphorylation of p38 and ERK1/2 and the phosphorylation and degradation of IκB-α in DCs (Figure 1D). We next confirmed the activation of MAPKs and NF-κB during Rv2005c-induced DC activation by using specific pharmacological inhibitors, including a p38 inhibitor (SB203580), an ERK1/2 inhibitor (U0126), a JNK inhibitor (SP600125), and an NF-κB inhibitor (Bay 11-0782). All the inhibitors significantly suppressed the Rv2005c-induced production of proinflammatory cytokines by BMDCs (Figure 1E). Although the endotoxin content was continuously monitored during the preparation of the recombinant protein, we further assessed LPS contamination by proteinase K treatment or heat denaturation, which suppressed the ability of Rv2005c to trigger DC maturation (Appendix A). The effects of Rv2005c were not inhibited by polymyxin B treatment, whereas those of LPS were significantly inhibited by polymyxin B treatment. These results suggest that Rv2005c induces DC maturation and activation via the MAPK and NF-κB signaling pathways. 

### 3.2. T Cells Activated by Rv2005c-Matured DCs Do Not Inhibit Intracellular Mtb Growth

Increased expression of surface molecules on BMDCs promotes their interaction with and activation of T cells. To precisely characterize the effects of Rv2005c on the interactions between BMDCs and T cells, we performed a syngeneic MLR assay using OT-II TCR transgenic CD4^+^ T cells. The proliferation of the T cells that were cocultured with Rv2005c-treated BMDCs pulsed with OVA_323-339_ was compared to that of the T cells cocultured with untreated BMDCs pulsed with OVA peptides (Figure 2A). Furthermore, OT-II TCR transgenic CD4^+^ T cells cocultured with Rv2005c-treated BMDCs pulsed with OVA peptides produced significantly higher levels of IFN-γ and IL-2, but not IL-4, than those primed with untreated BMDCs or LPS-treated DCs (Figure 2B). Next, we tested whether Rv2005c-matured DCs could actually play a role in the control of Mtb. Pharmacologically activated T cells were cocultured with Rv2005c-matured DCs for 72 h and they were added to Mtb-infected bone marrow-derived macrophages (BMDMs). As shown in Figure 2C, T cells activated by Rv2005c-matured DCs showed some bacterial growth inhibition, this inhibition was not significantly different from that of T cells activated by untreated DCs or infection control. Nevertheless, T cells activated by Rv2005c-matured DCs induced significant IFN-γ and IL-2 production, but not IL-4 production, by Mtb-infected macrophages compared to T cells activated by untreated DCs (Figure 2D).

### 3.3. Increased Expression of Rv2005c by Mtb Within Macrophages under Hypoxic Conditions Induces Intracellular Bacterial Growth Inhibition by T Cells Activated by Rv2005c-Matured DCs

Rv2005c is one of the Mtb genes induced by hypoxia [26] and is also upregulated during dormancy and reactivation [35]. Therefore, we investigated whether Rv2005c could be expressed in the intracellular environment of macrophages treated with CoCl_2_, which induces hypoxic conditions [36]. First, we determined the expression of HIF-1α and HspX (Rv2031c) by Mtb-infected BMDMs cultured with different concentrations of CoCl_2_ for 24 h. HIF-1α and HspX are typical proteins expressed by eukaryotic cells and Mtb, respectively, under hypoxic conditions [36,37]. Immunoblot assays showed that the expression of both proteins was clearly induced in macrophages treated with 300 μM CoCl_2_ (Figure 3A). Rv2005c was weakly expressed in Mtb-infected macrophages without CoCl_2_ treatment, but its expression was increased by CoCl_2_ treatment in a dose-dependent manner (Figure 3B), indicating that Rv2005c expression by Mtb was enhanced under stress conditions. There was no difference in ESAT-6 expression, which is a T cell-stimulating Mtb antigen, after CoCl_2_ treatment. Next, we examined whether CoCl_2_ was cytotoxic in BMDMs. As shown in Figure 3C, CoCl_2_ (300 μM) displayed no cellular toxicity in BMDMs regardless of Mtb infection and/or Rv2005c treatment, indicating that CoCl_2_ alone did not affect cell survival in our assay system. 

We repeated the same experiments shown in Figure 2C under CoCl_2_ treatment conditions. First, to test whether CoCl_2_ itself could inhibit Mtb growth, Mtb infected macrophages were treated with various concentrations (0–1000 μM) of CoCl_2_. As shown in Appendix A, 1000 μM CoCl_2_ slightly inhibited Mtb growth, but 300 μM CoCl_2_ did not inhibit Mtb growth. Next, T cells activated by Rv2005c-matured DCs did not inhibit the growth of Mtb in macrophages, but compared to their effects in the infection control or normoxic conditions, these T cells significantly inhibited the bacterial growth when the Mtb-infected macrophages were treated with CoCl_2_ for 24 h before the addition of the T cells (Figure 3D). Under the same conditions, T cells activated by Rv2005c-matured DCs induced significantly higher IFN-γ and IL-2 production by Mtb-infected macrophages pretreated with CoCl_2_ than by Mtb-infected macrophages under normoxic conditions (Figure 3E). These results suggest that macrophages present the Rv2005c peptide expressed by Mtb under hypoxic conditions on their surfaces and then interact with Rv2005c-specific T cells activated by DCs. 

### 3.4. Rv2005c Does Not Exhibit Significant Vaccine Efficacy 

It has been reported through comprehensive literature and in silico-based analyses that Rv2005c is one of the 45 top-ranking antigens for vaccine studies [29]. Therefore, we tested the direct vaccine effect of Rv2005c in a mouse model. The mice were immunized with Rv2005c/MPL-DDA three times and challenged with Mtb six weeks after the final immunization (Appendix A). Six weeks after challenge, the bacterial burden in the lungs was significantly reduced in the BCG-injected group compared to the infection control group, but the bacterial burden was not reduced in the Rv2005c-immunized group (Appendix A), although Rv2005c-specific IFN-γ production was observed in lung cells in the Rv2005c-immunized group (Appendix A). We next determined whether boosting BCG with Rv2005c could improve the protective efficacy. The Rv2005c-boosted mice exhibited a slightly lower bacterial load in the lung than the BCG-vaccinated mice, but this difference was not significant, although Rv2005c-specific IL-17 and IFN-γ production before the challenge and Rv2005c-specific IL-2 and IFN-γ production after the challenge were observed in the Rv2005c-boosted mice (Appendix A). Furthermore, we investigated the adjunctive immunotherapeutic potential of Rv2005c in a mouse model that was established in our lab with the Mtb H37Ra strain. The mice were infected with Mtb (1 × 10^6^ CFU/mouse), and three weeks postinfection, the mice were treated with INH and RIF for four weeks and immunized three times with Rv2005c/MPL-DDA during and after treatment, and the bacterial burden in the lungs was measured at the indicated time (Appendix A). The bacterial numbers in the lungs were maintained with a slight decrease until 16 weeks postinfection, but chemotherapy resulted in an undetectable bacterial number in the lungs seven weeks postinfection, at which time chemotherapy was terminated (Appendix A). However, the bacteria rapidly regrew in the lungs by nine weeks after the termination of chemotherapy, and the bacterial numbers in the lungs of these mice were similar to those in the lungs of the untreated mice. Unexpectedly, no inhibition of bacterial regrowth was observed in the Rv2005c-immunized mice (Appendix A). 

### 3.5. Intracellular Mtb Growth Inhibition by T Cells Activated by Rv2882c-Rv2005c Fusion Protein-Matured DCs Is Enhanced by Hypoxic Conditions

As described above, Rv2005c alone could not induce any vaccine efficacy in the mouse model, and it has been reported that this protein is expressed during dormancy and reactivation [35]. Therefore, to generate a therapeutic vaccine based on Rv2005c, we fused Rv2005c to the macrophage-activating protein Rv2882c that exerts a BCG-boosting effect [22]. The recombinant his-tagged Rv2882c-Rv2005c fusion protein was purified from *E. coli* sonicated extracts. The purified fusion protein showed a major single band at 50 kDa by SDS-PAGE (Figure 4A). Next, we tested whether the fusion protein could activate DCs. The Rv2882c-Rv2005c-stimulated BMDCs secreted significant levels of TNF-α, IL-1β and IL-12p70 compared to the untreated BMDCs (Figure 4B). Even at a concentration of 0.5 μg/mL, the fusion protein showed activity that was comparable to that of LPS (100 ng/mL). T cells activated by fusion protein-matured DCs produced significantly higher levels of IFN-γ, IL-2 and IL-17 than those activated by untreated DCs or LPS-treated DCs (Figure 4C). The endotoxin content was monitored throughout the preparation of the recombinant protein, and we further confirmed that the effects of Rv2005c were not inhibited by pretreatment with polymyxin B (Appendix A). Finally, we tested whether T cells activated by Rv2882c-Rv2005c-matured DCs could inhibit Mtb growth. We performed the same experiment as shown in Figure 3D. As shown in Figure 4D, T cells activated by Rv2882c-Rv2005c-matured DCs significantly inhibited Mtb growth in macrophages compared to infection control or T cells activated by untreated DCs; in addition, T cells activated by Rv2882c-Rv2005c-matured DCs more significantly inhibited the bacterial growth when the Mtb-infected macrophages were exposed to CoCl_2_ before the addition of the T cells compared to their inhibition of the bacterial growth in macrophages under normoxia conditions. These results suggest that the Rv2882c fusion enhances the antimycobacterial activity of Rv2005c.

### 3.6. Rv2882c-Rv2005c Protein Shows Significant Immunotherapeutic Effects 

We evaluated the therapeutic effect of Rv2882c-Rv2005c in our established mouse model. The mice were infected with Mtb (1 × 10^4^ CFU/mouse), treated with INH and RIF, and immunized three times with Rv2882c-Rv2005c/MPL-DDA or MPL-DDA adjuvant alone. The bacterial burden in the lungs was measured at the indicated time (Figure 5A). Mtb was not detected in the lungs of the Rv2882c-Rv2005c-immunized mice after three weeks of treatment (six weeks postinfection), but Mtb was still growing in the MPL-immunized mice (Figure 5B). At nine weeks postinfection (three weeks after the termination of chemotherapy), Mtb did not grow in the lungs of either group (Figure 5B). The bacteria rapidly regrew in the lungs of the MPL-immunized mice by 12 weeks and 20 weeks postinfection, but the bacterial growth was significantly suppressed in the Rv2882c-Rv2005c-immunized mice (Figure 5B). These findings suggest that Rv2882c-Rv2005c immunotherapy might be effectively used as an adjunct to chemotherapy. The cytokines in the culture supernatants of the lung cells restimulated with Rv2882c-Rv2005c at 6, 9, 12, and 20 weeks postinfection were measured. The production of IFN-γ, IL-2, and IL-17 was significantly higher in the cells from the Rv2882c-Rv2005c-immunized mice and peaked at 12 weeks postinfection, which was two weeks after the final immunization (Figure 5C). We also evaluated the frequency of multifunctional T cells after in vitro stimulation with Rv2882c-Rv2005c (Figure 6A) at the same time points. The numbers of Ag-specific IFN-γ^+^IL-2^+^ T cells were increased two weeks (six weeks postinfection) after the first immunization of the mice with adjuvant or Rv2882c-Rv2005c (Figure 6A). At nine weeks postinfection, when Mtb was not growing, the numbers of Ag-specific IFN-γ^+^ T cells increased. At 12 and 20 weeks, the numbers of multifunctional T cells, including IFN-γ^+^TNF-α^+^IL-2^+^, IFN-γ^+^IL-2^+^, IFN-γ^+^TNF-α^+^, and TNF-α^+^IL-2^+^ T cells, were significantly increased in the Rv2882c-Rv2005c immunized group compared to other groups and peaked at 12 weeks postinfection (Figure 6B). The numbers of IFN-γ^+^TNF-α^+^IL-2^+^ T cells were the highest among the multifunctional T cells in the Rv2882c-Rv2005c immunized mice at 12 weeks post infection (Figure 6B), and the numbers of IFN-γ^+^TNF-α^+^ T cells were sustained until 20 weeks postinfection. 

## 4. Discussion

Several TB vaccines are in clinical trials, but only a few antigens, such as Ag85 or ESAT-6, have been used in these vaccines [23]. There is accumulating evidence that the inclusion of latency-associated Ags, which are specifically encoded by the DosR regulon, will be important in the development of a more potent TB vaccine [38]. For example, H56 and ID93, the most advanced multistage subunit vaccines in clinical trials, contain the latency-associated Ags Rv2660c and Rv1813, respectively [18,39]. Therefore, the discovery and selection of target antigens shown immunogenicity and protective efficacy in preclinical animal models are required for the development of an improved TB vaccine. It has been reported that the expression of the Rv2005c gene in hypoxic conditions is regulated by DosR [40], and the Rv2005c protein is upregulated during the dormancy and reactivation of Mtb [35]. However, there is no report about the immunoreactivity of Rv2005c, which is a predicted vaccine candidate. Therefore, we investigated whether Rv2005c had potential as a vaccine, and the Rv2005c protein was first identified as a seroreactive antigen in TB patients. We found that T cells activated by Rv2005c-matured DCs triggered antimycobacterial activity in hypoxic macrophages, but the single Rv2005c protein did not exhibit significant efficacy as a vaccine (prophylactic, pre-exposure, or post-exposure). Therefore, we established the hypothesis that fusion with the macrophage-activating antigen Rv2882c, which resulted in Mtb growth inhibition and short term vaccine efficacy, enhanced the immunotherapeutic activity of Rv2005c. 

Several mycobacterial proteins have been reported to induce DC maturation and Th1 or Th2 polarization [30,41]. In the present study, Rv2005c stimulated BMDCs to upregulate costimulatory molecules, such as CD40, CD80, CD86, and MHC II molecules and induced Th1 responses. However, T cells activated by Rv2005c-matured DCs did not affect the growth of Mtb in macrophages (Figure 2C). Because this result was possibly due to the lower expression of Rv2005c by intracellular Mtb, we treated Mtb-infected macrophages with CoCl_2_ to induce a hypoxic-mimetic microenvironment [42], which was proven by HIF-1α and Mtb HspX expression. Interestingly, Rv2005c expression in macrophages was enhanced by CoCl_2_ treatment, and T cells activated by Rv2005c-matured DCs inhibited the growth of Mtb in the macrophages treated with CoCl_2_ (Figure 3D). It is well known that Mtb rapidly expresses HspX under stress conditions, such as hypoxia and latency, and HIF-1α is also expressed in cells exposed to hypoxia [36,37]. CoCl_2_ has been used to stabilize the transcription factor HIF-1α and to mimic hypoxia [36,42]. These results suggest that under hypoxic conditions, a peptide derived from Rv2005c is presented on the macrophage surface by MHC class II molecules and then is recognized by activated Rv2005c-specific T cells. We hypothesized that Rv2005c can also be expressed by Mtb located in granulomas, which contain a hypoxic microenvironment. However, Rv2005c did not show vaccine efficacy, BCG-prime boosting effects, or therapeutic efficacy in our mouse model (Appendix A). It is possible that these results occurred because our mouse model does not form necrotic granulomas resembling those observed in human TB and because we used the less virulent Mtb H37Ra instead of H37Rv for the mouse infection experiments. Our data and other reports [43] suggest that compared with H37Rv, H37Ra forms fewer and smaller granulomas in the peribronchial area of the lungs of mice. 

We previously identified Rv2882c, which strongly induces macrophage activation and exhibits BCG boosting efficacy [22]. This protein is a ribosomal recycling factor and is found in the membrane fraction [22]. Therefore, we fused Rv2005c, a latency-associated protein, to Rv2882c, which is expressed in the replication stage of Mtb. In the present study, we found that the Rv2882c-Rv2005c fusion protein induced the maturation of BMDCs even at low concentrations and that the growth of Mtb in macrophages under normoxic conditions was significantly inhibited by T cells activated by Rv2882c-Rv2005c-matured DCs (Figure 4D) but not by T cells activated by Rv2005c-matured DCs (Figure 3D). The Mtb growth inhibition induced by T cells activated by Rv2882c-Rv2005c-matured DCs was significantly enhanced when the macrophages were exposed to hypoxia (Figure 4D). These results indicate the synergistic effects of the fusion of these proteins on the antimycobacterial activity in vitro.

Some DNA and subunit vaccines have shown promising therapeutic efficacy in conjunction with conventional chemotherapy against TB [18,44]. Vaccination during and/or after chemotherapy can shorten the duration of chemotherapy and inhibit the regrowth of Mtb after chemotherapy in a mouse model. In this study, we established a mouse model using the less virulent Mtb H37Ra to evaluate the therapeutic effects of protein vaccines. We observed that Mtb H37Ra was cleared in the mice treated with chemotherapy and then regrew after the discontinuation of chemotherapy. Compared with adjunctive immunotherapy with the adjuvant alone, adjunctive immunotherapy with Rv2882c-Rv2005c significantly reduced the bacterial load in the lungs and significantly inhibited the bacterial regrowth (Figure 5), indicating the immunotherapeutic potential of Rv2882c-Rv2005c as an adjunct to chemotherapy in preventing the reactivation of latent TB. There are no biomarkers correlated with TB vaccine-induced protection. The multifunctional CD4^+^ T cells have been shown to provide a good marker of protection against TB in animal models [45]. In this study, Rv2882c-Rv2005c adjunctive immunotherapy resulted in a significantly higher frequency of vaccine-specific multifunctional T cells among the lung lymphocytes, and this frequency was maintained even at 10 weeks after the final immunization (Figure 6). Hoang et al. reported that vaccine-specific, multifunctional CD4^+^ T cells are similarly increased at 17 weeks postinfection in the lungs of mice immunized with H56 and H28 in a model of postexposure vaccination; however, postexposure vaccination with H56 but not with H28 efficiently inhibited the bacterial regrowth assessed at 44 weeks postinfection [46]. Immunotherapeutic studies using alpha-crystalline-based DNA vaccines indicated that there is no significant difference in the multifunctional splenic T cells among vaccine types, but expansion of effector memory T cells is associated with immunotherapeutic effects of DNA vaccination [47]. Taken together, these studies suggest that the identification of biomarkers correlated with the efficacy of adjunctive immunotherapy is required.

In conclusion, although further investigation of the therapeutic efficacy of our fusion protein against the virulent Mtb H37Rv and examination of the detailed immune mechanism are needed, our data suggest that the Rv2882c-Rv2005c fusion protein should be a candidate for the rational design of an immunotherapy vaccine.

## 5. Conclusions

Collectively, our study demonstrated that the Rv2882c-Rv2005c fusion protein was enhanced antimycobacterial effect in vitro/in vivo, and induced antigen-specific multifunctional CD4^+^ T cells in Mtb infected mice lung. Like this, construction of fusion protein and validation of immunotherapeutic efficacy will be used as a basis to effectively improve the therapeutic vaccines against tuberculosis.

## Figures and Tables

**Figure 1 vaccines-08-00370-f001:**
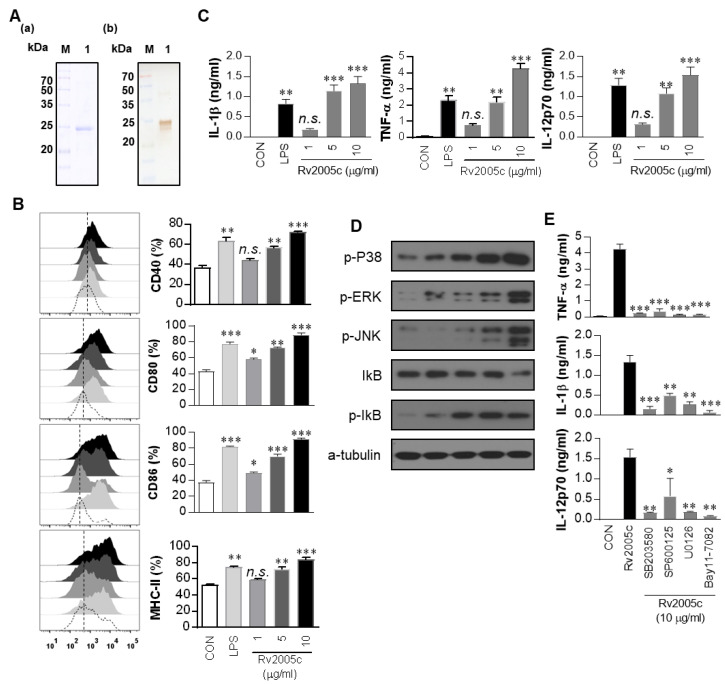
Recombinant Rv2005c induces dendritic cell (DC) maturation through the MAPK pathway. (**A**) The purified, recombinant Rv2005c protein was subjected to (**A**) Coomassie blue staining sodium dodecyl sulfate polyacrylamide gel electrophoresis (SDS-PAGE) and (**B**) western blot analysis using a mouse anti-His Ab. DCs were activated with the indicated concentration of Rv2005c or LPS (100 ng/mL) for 24 h. (**B**) Activated DCs were stained with anti-CD40, anti-CD80, anti-CD86, or anti-MHC class II Ab, and the expression of these surface markers was analyzed. The median fluorescence intensity (MFI) of the positive cells is shown for each panel. The bar graphs show the mean ± SEM (*n* = 5). (**C**) The TNF-α, IL-1β, and IL-12p70 levels in the culture medium were measured by ELISA. The data are presented as the mean ± SEM (*n* = 5). (**D**) The protein production by DCs treated with Rv2005c for the indicated periods was analyzed by immunoblotting using each specific Ab: phospho-p38 (p-p38), p38, phospho-ERK1/2 (p-ERK1/2), phospho-IκB-α, and IκB-α. (**E**) DCs were pretreated with pharmacological inhibitors of p38 (SB203580, 20 μM), ERK1/2 (U0126, 10 μM), JNK (SP600125, 20 μM), Bay11-7082 (20 μM), or DMSO (vehicle control) for 1 h prior to treatment with 10 μg/mL Rv2005c protein for 24 h. The cytokine levels in the culture supernatants were measured by ELISA. The data shown are the mean ± SEM; * *p* < 0.05, ** *p* < 0.01, or *** *p* < 0.001 for the inhibitor-treated samples compared to the Rv2005c-treated controls. n.s.: no significant difference.

**Figure 2 vaccines-08-00370-f002:**
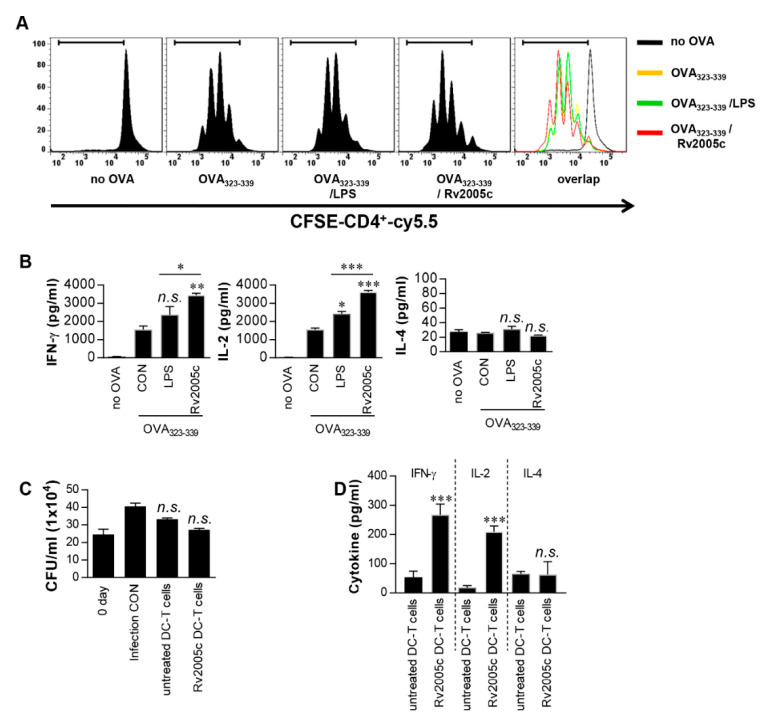
T cells activated by Rv2005c-matured DCs do not inhibit intracellular Mtb growth. (**A**) Transgenic OVA-specific CD4^+^ T cells were isolated, stained with CFSE, cocultured for 96 h with DCs treated with Rv2005c (10 μg/mL) or LPS (100 ng/mL), and pulsed with OVA_323–339_ (1 μg/mL) for OVA-specific CD4^+^ T cells. T cells alone and T cells cocultured with untreated DCs served as the controls. The proliferation of the OT-II^+^ T cells was then assessed by flow cytometry. (**B**) The culture supernatants were harvested after 24 h, and IFN-γ, IL-2, and IL-4 were analyzed by ELISA. All the data were expressed as mean ± SEM. The levels of significance (* *p* < 0.05, ** *p* < 0.01, or *** *p* < 0.001) for the treated samples compared to the appropriate controls (T cell/OVA_323–339_ pulsed DCs). n.s.: no significant difference. (**C**,**D**) Pharmacologically activated T cells cocultured with unstimulated DCs or Rv2005c-stimulated DCs at a DC:T cell ratio of 1:10 for 3 days, and then these T cells were cocultured with BMDMs infected with Mtb. (**C**) The intracellular Mtb growth in the BMDMs was determined at time point 0 (0 days) and 3 days after coculture with T cells. (**D**) The quantities of cytokine levels in the culture supernatants were determined by ELISA. All the data were expressed as mean ± SDs (*n* = 3). The levels of significance (* *p* < 0.05, ** *p* < 0.01, or *** *p* < 0.001) for the BMDMs cocultured with T cells compared to the control BMDMs. n.s.: no significant difference.

**Figure 3 vaccines-08-00370-f003:**
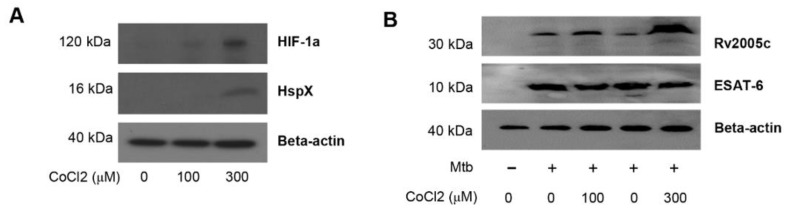
Increased expression of Rv2005c by Mtb within bone marrow-derived macrophages (BMDMs) under hypoxic conditions induces Mtb growth inhibition by T cells activated by Rv2005c-matured DCs. (**A**) BMDMs were infected with H37Rv at MOI = 5 for 2 h, and then, the extra bacteria were removed by washing. Then, the cells were treated with 100–300 μM CoCl_2_ for 8 h and lysed, and the total cell lysates were separated by SDS–PAGE, followed by immunoblot analysis using antibodies against HIF-1α and HspX. HspX was the Mtb hypoxia control. (**B**) BMDMs were infected with/without H37Rv at MOI = 5 for 2 h, and then, the extra bacteria were removed by washing. Then, the cells were treated with/without 100–300 μM CoCl_2_ for 8 h and lysed, and the total cell lysates were separated by SDS–PAGE, followed by immunoblot analysis using antibodies against Rv2005c and ESAT-6. (**C**) BMDCs were cultured for 24 h in the presence of 300 μM CoCl_2_ and analyzed by flow cytometry. STS (500 nM) was used as the control. BMDMs infected with or without Mtb were stained with anti-CD11b, annexin V, and PI. The percentage of positive cells (annexin V- and PI-stained cells) in each quadrant is indicated. The results are representative of three experiments. (**D**,**E**) Pharmacologically activated T cells or T cells activated by unstimulated DCs or Rv2005c-stimulated DCs at a DC:T cell ratio of 1:10 for 3 days were cocultured with BMDMs infected with Mtb under normoxic (N) or hypoxia (H) conditions. (**D**) The intracellular Mtb growth in the BMDMs was determined at time point 0 (0 days) and 3 days after coculture with T cells or without T cells (control). (**E**) The quantities of cytokine levels in the culture supernatant were determined by ELISA. All the data were expressed as mean ± SDs (*n* = 3). The levels of significance (* *p* < 0.05, ** *p* < 0.01, or *** *p* < 0.001, **** *p* < 0.0001) for the BMDMs cocultured with T cells compared to the control BMDMs. n.s.: no significant difference.

**Figure 4 vaccines-08-00370-f004:**
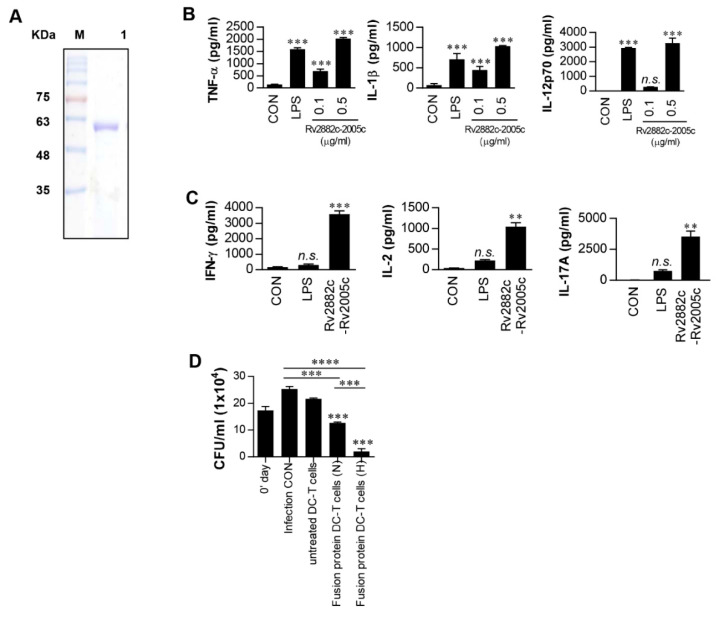
Intracellular Mtb growth inhibition by T cells activated by Rv2882c-Rv2005c fusion protein-matured DCs is enhanced under hypoxic conditions. (**A**) The purified, recombinant Rv2882c-Rv2005c protein was analyzed by SDS-PAGE with Coomassie blue staining. DCs were activated with the indicated concentration of Rv2882c-Rv2005c or LPS (100 ng/mL) for 24 h. (**B**) The TNF-α, IL-1β, and IL-12p70 levels in the culture medium were measured by ELISA. The data are presented as the mean ± SEM (*n* = 5). (**C**) T cells were activated by unstimulated DCs, LPS-stimulated DCs, or Rv2882c-Rv2005c-stimulated DCs at a DC:T cell ratio of 1:10 for 3 days, and the cytokine levels were determined. The data were expressed as mean ± SDs (*n* = 3) The levels of significance (* *p* < 0.05, ** *p* < 0.01, or *** *p* < 0.001) for the BMDMs cocultured with T cells compared to the control BMDMs. n.s.: no significant difference. (**D**) Pharmacologically activated T cells or T cells activated by unstimulated DCs or Rv2882c-Rv2005c-stimulated DCs at a DC:T cell ratio of 1:10 for 3 days were cocultured with BMDMs infected with Mtb under normoxic (N) or hypoxic (H) conditions. The intracellular Mtb growth in the BMDMs was determined at time point 0 (0 days) and 3 days after coculture with T cells or without T cells (control). The data were expressed as mean ± SDs (*n* = 3). The levels of significance (* *p* < 0.05, ** *p* < 0.01, *** *p* < 0.001, or **** *p* < 0.0001) for the BMDMs cocultured with T cells compared to the control BMDMs. n.s.: no significant difference.

**Figure 5 vaccines-08-00370-f005:**
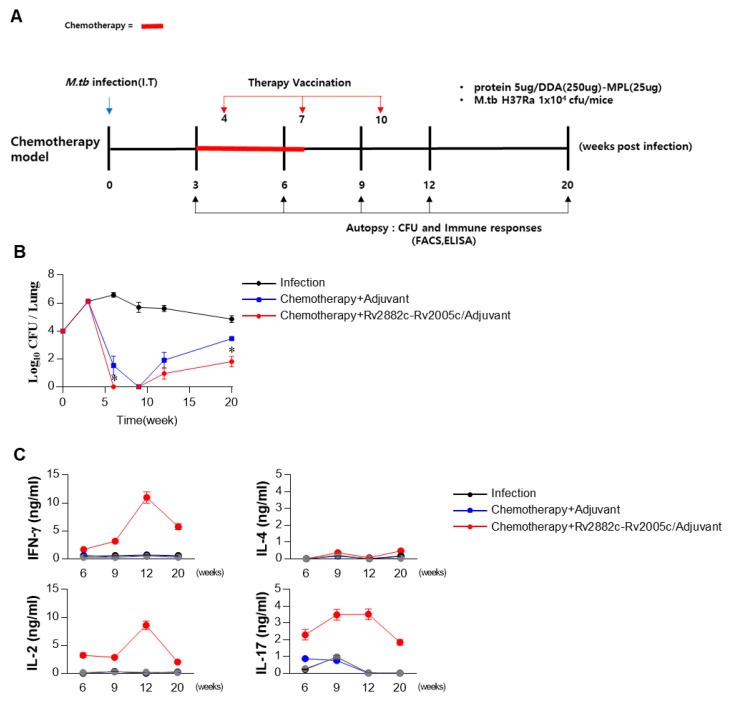
The Rv2882c-Rv2005c-specific cytokine shortens the duration of chemotherapy and inhibits Mtb reactivation. (**A**) Schematic diagram in each panel shows the detailed treatment schedule of the drug-treated and drug + immunotherapy-treated groups. (**B**) The bacterial loads in the lungs of mice following different treatment schedules compared to those in the control untreated mice. Chemotherapy was given for 4 weeks. The bacterial counts are expressed as the log_10_ value of the colony forming units. The data represent the mean CFU of five mice in each group. The mean ± SEM (*n* = 3) and statistical significance (* *p* < 0.05) are indicated for the treated groups compared to the adjuvant control groups. (**C**) Lung cells (2 × 10^6^/well) were treated with Rv2882c-Rv2005c (5 μg/mL) for 24 h. The levels of cytokines in the culture supernatants were determined by ELISA.

**Figure 6 vaccines-08-00370-f006:**
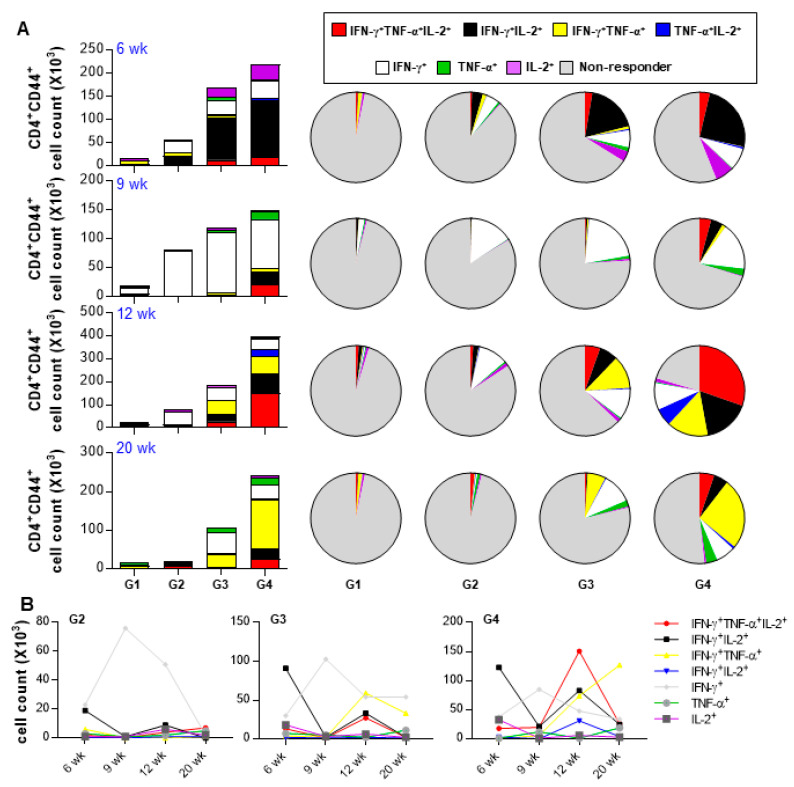
Analysis of polyfunctional cytokine production by Rv2882c-Rv2005c-specific CD4^+^ T cells in infected lungs after treatment. The percentages of activated multifunctional CD4^+^CD44^+^CD62^−^ T cells in the drug- and drug plus Rv2882c-Rv2005c-treated groups were evaluated at different time points during the course of therapy. The percentage of the respective cells in the infected control group is shown in the inset of the bar diagram. G1 = naïve G2 = Infection only G3 = Chemotherapy + adjuvant G4 = Chemotherapy + Rv2882c-Rv2005c / adjuvant. Data are expressed as (**A**) Pie chart, (**B**) Linear graph.

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
