# Peer review of "Mycobacterium tuberculosis Rv2005c Induces Dendritic Cell Maturation and Th1 Responses and Exhibits Immunotherapeutic Activity by Fusion with the Rv2882c Protein"

_vaccines, 2020, doi:10.3390/vaccines8030370_

Round 1

Reviewer 1 Report

Mycobacterium tuberculosis Rv2005c induces dentritic cell maturation and Th1 responses and exhibits immunotherapeutic activity by fusion with Rv 2882c protein 

Dear author and editor:

This is an interesting study talked about fused protein Rv2005c-Rv2882c of Mycobacterium tuberculosis and suggested it as a candidate for immunotherapeutic vaccine. The work is well written and the experimental art is solid. 

The article could be published in Vaccines after a minor revision. 

I have some comments on the work:

  • In the results part, i think it is better to merge 3.1 and 3.2 results in one first results saying that the recombinant protein induces maturation and activation of DCs. (the identification and preparation of Rv2882c protein are related to material and methods).
  • Did you reached a purified protein 100% free of endotoxins?
  • Is there any effect of the Rv2882c on macrophages activation in the literature or on alveolar epithelial cells?
  • The author used immunoblot to confirm the protein expression. why did you not use RT-PCR which is preferable for expression data and you can give your data as an increase in fold expression?
  • The hypoxia itself increase the macrophages activity against Mtb. how do you know that the observed effect related to protein and not related to the hypoxic activation of macrophages. 
  • Did you try to verify the Mtb growth inhibition under hypoxic condition without treating the cells with the fused protein?
  • The aim from using this fused protein is not clear in the text.
  • In the figure 5, the author used the same title of the results. i prefer to use a figure title different from the results title. 
  • why did you not try to immunize the untreated mice to verify the effect of the vaccine alone without chemotherapy which significantly reduced the bacterial load in the lung of mice after 5 weeks of infection?

Thank you very much, best regards 

Author Response

[Reviewer 1]

Dear author and editor:

This is an interesting study talked about fused protein Rv2005c-Rv2882c of Mycobacterium tuberculosis and suggested it as a candidate for immunotherapeutic vaccine. The work is well written and the experimental art is solid.

The article could be published in Vaccines after a minor revision.

I have some comments on the work:

[Q1] In the results part, i think it is better to merge 3.1 and 3.2 results in one first results saying that the recombinant protein induces maturation and activation of DCs. (the identification and preparation of Rv2882c protein are related to material and methods).

[A1] We appreciate for your excellent comment. We merged results of 3.1 and 3.2, and rephrased the title to “The recombinant Rv2005c protein induces maturation and activation of DCs”.

[Q2] Did you reached a purified protein 100% free of endotoxins?

[A2] We completely understand the concerns raised by the reviewer. In the final step for purification of the recombinant proteins, all purified proteins were passed through Polymyxin B-agarose column to remove endotoxin. In addition, as your concerns, we have performed the LAL assay to determine endotoxin concentration and the data were added in results section (page 9, Line 326). For your reference, we have provided the LAL data as the follow:

Reference Figure 1. Confirmation of endotoxin decontamination from purified Rv2005c and Rv2882c-Rv2005c. The amount of residual LPS in the Rv2005c or Rv2882c-Rv2005c preparation was estimated using the Limulus amoebocyte lysate (LAL) test according to the manufacturer’s instructions.

[Q3] Is there any effect of the Rv2882c on macrophages activation in the literature or on alveolar epithelial cells?

[A3] We appreciate your excellent comment. We did not analyze any effect of Rv2882c on alveolar epithelial cell, and have focused on bone marrow derived macrophage activation because we have previously reported that Rv2882c induces activation of macrophages through TLR4 and exhibits vaccine potential (PMID: 27711141). There is no any other report about Rv2882c up to date.

[Q4] The author used immunoblot to confirm the protein expression. why did you not use RT-PCR which is preferable for expression data and you can give your data as an increase in fold expression?

[A4] We have initially analyzed Rv2005c expression by RT-PCR. Expression of Rv2005c mRNA significantly increased in Mtb-infected macrophage under hypoxia than normoxia. The same results were obtained in the repeated experiments, but the data quality was very low, and we have considered those as the preliminary results. We actually thought that determination of protein level is more important. For your reference, we have provided the RT-PCR data as the follow:

Reference Figure 2. Representative RT-PCR images showing expression of genes encoding Rv2005c, GAPDH and 16s rRNA in Mtb-infected macrophages. Mtb-infected macrophages at MOI 1 were incubated in presence or absence of CoCl2 and collected at 8 h. Lane 1; non-infected macrophages, lane 2; Mtb-infected macrophages, lane 3; Mtb-infected macrophages incubated in presence of CoCl2 100 μM, lane4; Mtb-infected macrophages incubated in presence of CoCl2 300 μM.

[Q5] The hypoxia itself increase the macrophages activity against Mtb. how do you know that the observed effect related to protein and not related to the hypoxic activation of macrophages. Did you try to verify the Mtb growth inhibition under hypoxic condition without treating the cells with the fused protein?

[A5] We thank the referee for the excellent comments and suggestions, which are extremely valuable for improving the quality of our manuscript. First, we did not confirm the activity of macrophage when treating macrophage with CoCl2. Second, we apologize for our mistake of missing Mtb growth inhibition data in hypoxia condition. We have added CFU data in supplementary figure 3 and described in Results section (Page 13, Line 435-438). Briefly, there were no any effect of CoCl2 on the intracellular bacterial growth up to 500 μM of CoCl2, and the bacterial growth was inhibited when treated 1000 μM CoCl2 in Mtb-infected macrophages. Based on this result, it suggests that CoCl2 300 μM did not affect the protein-mediated Mtb growth inhibition at least.

Supplementary Figure 3. Confirmation of Mtb growth inhibition by CoCl2. BMDMs were infected with H37Rv at MOI = 5 for 2 h, and then, the extra bacteria were removed by washing. Then, the cells were treated with 100 - 1000 μM CoCl2 for 8 h and lysed, then the intracellular Mtb growth in the BMDMs was determined at time point 0 (0 days) and 3 days after treat CoCl2. All the data were expressed as mean ± SDs (n = 3). The levels of significance (*p < 0.05) for the BMDMs treated with CoCl2 compared to the not treated with CoCl2. n.s.: no significant difference.

[Q6] The aim from using this fused protein is not clear in the text.

[A6] Thank you for your valuable indication. We have added the more detail rationale of making fusion protein in the revised manuscript as follow: We found that T cells activated by Rv2005c-matured DCs induce antimycobacterial activity in hypoxic macrophages, but Rv2005c single protein dose not exhibit significant vaccine (prophylactic, pre-exposure, post-exposure) efficacy. So, we established hypothesis that fusion with the macrophage-activating antigen Rv2882c, inducing Mtb growth inhibition in early stage vaccine efficacy, enhances the immunotherapeutic activity of Rv2005c. Therefore, chemotherapy was conducted to confirm the initial reduce bacteria burden and inhibit reactivation long term stage. We have described this point in Discussion section (Page 21, Line 624-631)

[Q7] In the figure 5, the author used the same title of the results. i prefer to use a figure title different from the results title.

[A7] We apologize for our mistake. We rephrased the figure title different from the results title.

[Q8] why did you not try to immunize the untreated mice to verify the effect of the vaccine alone without chemotherapy which significantly reduced the bacterial load in the lung of mice after 5 weeks of infection?

[A8] We appreciate these excellent comments. We described chemotherapy alone in Figure 5, it seems to have caused confusion. Therefore, we represent to “chemotherapy+adjuvant” in Figure 5. In addition, the therapeutic efficacy of a vaccine alone without chemotherapy has not been tested in several papers (PMID: 15661388, 22891286, 21258338). Basically, it is well known that a protein alone is not able to induce a significant immune response. Also, since the Mtb immunotherapy model has not been standardized, we have designed the experiment with reference to these papers.

Reviewer 2 Report

The authors describe that a Mtb-derived Rv2005c protein acts as an antigen, triggering antibody-responses in TB patients, but also as an innate cell stimulus making it an exciting candidate for vaccine design. However, the study concept is not convincing and some immunological concepts are not correctly explained and require further investigation. Here, I describe my main concerns that I hope will be well received and may improve the conclusions that can be drawn from the presented data.

  1. Effect of Rv2005 on DCs

Stimulation of innate cells (DCs and macrophages) by a recombinant Mtb-derived protein Rv2005c as described by the authors is very unlikely as this would mean that the protein is recognized by a pathogen pattern recognition receptor (PPRs) or other receptor on the cell surface. Known structures that are triggering this innate immune activation by PPRs are pathogen-common components as bacterial lipids, lipoproteins, complex carbohydrates and pathogene-derived DNA and RNA structures. Proteins are not a commonly-described stimuli of innate responses and not included in the known mycobacteria components-recognizing PPRs (TLR-2,-4-,9, MannoseR, DCAR, DCSign, etc). The authors do not discuss a possible mechanism that would lead to the DC activation and maturation of Rv2005 and if it possesses any special properties (structure) to account of this action. The finding of innate cell activation of Rv2005c and the mechanism (including receptor identification) would be of great interest. Instead a contamination of their recombinant protein could be the cause of the DC activation. They exclude in several control experiments a LPS contamination but other bacterial components could be responsible, even their CB staining in figure 1A shows several bands that are not discussed and indicate other components to be present in their purification. It would be a confirmation of the protein's purity to show a MS analysis. To have a proof of the specificity of action of this particular protein, a purification and similar assay of another recombinant mycobacterial protein that is known to be not activating DCs in parallel might be helpful.

  1. Naïve T cell priming in vitro

The T cell repertoire within naïve T cells is very broad to be able to recognize more or less any possible encountered antigen (estimations of around 10^6 to 10^8). To use naïve T cells to detect priming to one specific peptide or antigen in a cell culture assay is therefore to my understanding not possible (you may have none or max very few cells recognizing the presented peptide per well). This is why these assays are performed with mice from transgenic mice expressing one T cell receptor (=all responding to the presented antigen) as the authors did in in figure 2A. An alternative is to use T cells from infected mice if one is interested in recall responses to an expanded T cell population (this antigen must be triggering expansion) to have substantial T cell numbers to detect responses. Therefore, all experiments claiming naïve T cell priming after DC co-culture and also the co-culture with infected macrophages can not be connected to specific T cell activation but may be due to bystander activation or other mechanisms. This needs to be discussed or experimental set up should be re-thought (my recommendation). T cells from transgenic mice expressing specific T cell receptors recognizing presented Rv2005c peptides would be required to perform the experiments and draw the conclusions the authors did.

  1. Post-exposure vaccination

The combination in the fusion of two mycobacterial proteins may not automatically have the separate effects of the action on the individual proteins as described by the authors. It could be considered that their recognizable epitopes could change and as their mode of action is not elucidated this step may be premature.

Regarding the different experimental set ups that mainly show no effects besides the post exposure vaccination, I would like to point out that this experiment may miss some controls that are comparing the effect of the fusion-adjuvant with BCG or other mycobacterial proteins-adjuvant as the comparison to the adjuvant only is very in favor of the fusion-adjuvant and may not be superior to other mycobacterial proteins nor BCG.

Author Response

Comments and Suggestions for Authors

The authors describe that a Mtb-derived Rv2005c protein acts as an antigen, triggering antibody-responses in TB patients, but also as an innate cell stimulus making it an exciting candidate for vaccine design. However, the study concept is not convincing and some immunological concepts are not correctly explained and require further investigation. Here, I describe my main concerns that I hope will be well received and may improve the conclusions that can be drawn from the presented data.

Effect of Rv2005 on DCs

[Q1] Stimulation of innate cells (DCs and macrophages) by a recombinant Mtb-derived protein Rv2005c as described by the authors is very unlikely as this would mean that the protein is recognized by a pathogen pattern recognition receptor (PPRs) or other receptor on the cell surface. Known structures that are triggering this innate immune activation by PPRs are pathogen-common components as bacterial lipids, lipoproteins, complex carbohydrates and pathogene-derived DNA and RNA structures. Proteins are not a commonly-described stimuli of innate responses and not included in the known mycobacteria components-recognizing PPRs (TLR-2,-4-,9, MannoseR, DCAR, DCSign, etc). The authors do not discuss a possible mechanism that would lead to the DC activation and maturation of Rv2005 and if it possesses any special properties (structure) to account of this action. The finding of innate cell activation of Rv2005c and the mechanism (including receptor identification) would be of great interest. Instead a contamination of their recombinant protein could be the cause of the DC activation. They exclude in several control experiments a LPS contamination but other bacterial components could be responsible, even their CB staining in figure 1A shows several bands that are not discussed and indicate other components to be present in their purification. It would be a confirmation of the protein's purity to show a MS analysis. To have a proof of the specificity of action of this particular protein, a purification and similar assay of another recombinant mycobacterial protein that is known to be not activating DCs in parallel might be helpful.

[A1] We completely understand the concerns raised by the reviewer. First, we have published that various Mtb antigens can activate innate immune cells (Macrophage or DC) (PMID: 21993523, 28193909, 27711141, 25907170, 30862819, 29736404). In addition, several other groups have reported that the Mtb antigens activate innate immune cells (PMID: 22415304, 25689444, 30809214). A wide variety of Mtb proteins activate innate cells through TLR2/4 (PMID: 21993523, 28193909, 25907170, 30862819, 22415304, 25689444, 29736404). We also reported that Rv2882c which is fusion partner for Rv2005c induces macrophage activation via interacting with TLR4. Therefore, we have preliminary tested whether Rv2005c could interact with TLR2 or TLR4, but the clear data to prove these interactions could not be obtained. By our experience, all Mtb protein antigens did not induce activation of macrophages or dendritic cells, and also a protein is able to induce activation of these cells without TLR2 or TLR4 involvement. Second, we have checked the protein by loading the gel, reflecting reviewer comment. It is shown that Rv2005c main band was disappeared when inactivated Rv2005c (Rv2005c treated with PK and boiled Rv2005c) load the gel.

Reference Figure 1. Confirmation of denatured Rv2005c protein. Rv2005c denatured by heating for 1 h at 100°C and digested with Proteinase K (10 mg/mL) for 1 h at 37°C, and analyzed by SDS-PAGE with Coomassie blue staining (M: marker, Line 1:Rv2005c, Line 2: denatured Rv2005c).

It is suggested that decrease cytokines producing cause disappeared Rv2005c main band. In addition, as answers to Reviewer 1, the issue about LPS contamination have been exclude by passing through Polymyxin B-agarose column for removing LPS, direct determination of endotoxin concentration by the LAL assay, and determination of immunologic activity of the protein with boiling, PK treatment, and PMP pretreatment.

Naïve T cell priming in vitro

[Q2] The T cell repertoire within naïve T cells is very broad to be able to recognize more or less any possible encountered antigen (estimations of around 10^6 to 10^8). To use naïve T cells to detect priming to one specific peptide or antigen in a cell culture assay is therefore to my understanding not possible (you may have none or max very few cells recognizing the presented peptide per well). This is why these assays are performed with mice from transgenic mice expressing one T cell receptor (=all responding to the presented antigen) as the authors did in in figure 2A. An alternative is to use T cells from infected mice if one is interested in recall responses to an expanded T cell population (this antigen must be triggering expansion) to have substantial T cell numbers to detect responses. Therefore, all experiments claiming naïve T cell priming after DC co-culture and also the co-culture with infected macrophages can not be connected to specific T cell activation but may be due to bystander activation or other mechanisms. This needs to be discussed or experimental set up should be re-thought (my recommendation). T cells from transgenic mice expressing specific T cell receptors recognizing presented Rv2005c peptides would be required to perform the experiments and draw the conclusions the authors did.

[A2] I totally agree with you. What we knowledge, CD28 antibody alone can activate memory or effector T cells, and CD3 agonistic signals are needed for activation of naive T cells. I'm sorry we didn't write down the experiment method in detail. The naïve T cells were pretreated with PMA (100 nM) and ionomycin (1000 nM) for pre-activation of Naïve T cell before co-culturing with protein-treated DCs (Page 7, line 242). Throughout our manuscript, we will correct naïve T cells as Pharmacologic activated T cells (PMA / Ionomycin activated T cells, P/I activated T cells). We have described it in Method section (page 7, Line 243)

Post-exposure vaccination

[Q3] The combination in the fusion of two mycobacterial proteins may not automatically have the separate effects of the action on the individual proteins as described by the authors. It could be considered that their recognizable epitopes could change and as their mode of action is not elucidated this step may be premature. Regarding the different experimental set ups that mainly show no effects besides the post exposure vaccination, I would like to point out that this experiment may miss some controls that are comparing the effect of the fusion-adjuvant with BCG or other mycobacterial proteins-adjuvant as the comparison to the adjuvant only is very in favor of the fusion-adjuvant and may not be superior to other mycobacterial proteins nor BCG.

[A3] We are very thankful to the reviewer for your deep comments. In the first, we have investigated the therapeutic effect of adjuvant alone as control in Figure 5 and supplementary Figure 6. Therefore, for clear understanding, we have changed the figure labeling as the follows; Chemotherapy à Chemotherapy + adjuvant, Chemotherapy + Rv2882c-Rv2005c à Chemotherapy + Rv2882c-Rv2005c/adjuvant for Fig 5; Chemotherapy à Chemotherapy + adjuvant, Chemotherapy + Rv2005c à Chemotherapy + Rv2005c/adjuvant for supplementary Figure 6. It is known that BCG has no an immunoadjunctive effect. In terms of other protein control, we already have included another fusion protein, which was performed simultaneously with Rv2882c-Rv2005c in our therapeutic model, but this fusion protein had no effect, and we could not provide the results of this protein in the manuscript because this protein is a novel and unpublished fusion protein. But, for your reference, we have provided the other control protein data (indicated as fusion X) as the follow:

Reference Figure 2. Effects of fusion-protein Immunotherapy on Lung CFU of Mtb infected of mice. The number of viable bacteria in the lungs of mice was determined 3, 6, 9, 12, and 20 weeks after infection.

Also, like your very good recommendation, we are investigating the efficacy of this fusion protein in a variety of ways (change of adjuvant, inoculation timing and inoculation route, or BCG boosting model etc.).

Round 2

Reviewer 2 Report

I thank the authors for their added explanations to my question 1. However, I miss answers to my comments “The authors do not discuss a possible mechanism that would lead to the DC activation and maturation of Rv2005 and if it possesses any special properties (structure) to account of this action.” And “It would be a confirmation of the protein's purity to show a MS analysis.”

I appreciate the corrections that in response to my question 2. However, I don’t see a change in the experiment starting in line 379 indicating that naïve CD4 T cells were primed and produced IFNg after co-culture with Rv2005-stimulated DCs. This is, as I wrote in my comment before, immunological not possible. (The T cell repertoire within naïve T cells is very broad to be able to recognize more or less any possible encountered antigen (estimations of around 10^6 to 10^8). To use naïve T cells to detect priming to one specific peptide or antigen in a cell culture assay is therefore to my understanding not possible (you may have none or max very few cells recognizing the presented peptide per well). This is why these assays are performed with mice from transgenic mice expressing one T cell receptor (=all responding to the presented antigen) as the authors did in in figure 2A. An alternative is to use T cells from infected mice if one is interested in recall responses to an expanded T cell population (this antigen must be triggering expansion) to have substantial T cell numbers to detect responses. Therefore, all experiments claiming naïve T cell priming after DC co-culture and also the co-culture with infected macrophages can not be connected to specific T cell activation but may be due to bystander activation or other mechanisms. This needs to be discussed or experimental set up should be re-thought (my recommendation). T cells from transgenic mice expressing specific T cell receptors recognizing presented Rv2005c peptides would be required to perform the experiments and draw the conclusions the authors did.)

I agree with the author’s answer to my question 3.

Author Response

[Q1] I thank the authors for their added explanations to my question 1. However, I miss answers to my comments “The authors do not discuss a possible mechanism that would lead to the DC activation and maturation of Rv2005 and if it possesses any special properties (structure) to account of this action.” And “It would be a confirmation of the protein's purity to show a MS analysis.”

[A1] We fully understand your concerns. As answered in 1st revision, we have already confirmed that the immunoreactivity of Rv2005c on DC was not due to LPS contamination. Denaturation of purified Rv2005c fraction by proteinase K treatment resulted in loss of bioactivity of Rv2005c, indicating suggest that activities of the purified recombinant protein were due to a protein. The purified recombinant Rv2005c showed a major band at 25-kDa on Coomassie blue stained gel (Fig. 1A), although the minor contaminated bands were also showed.

And also, as answered in 1st revision, we could not find that Rv2005c interact with a surface molecule such as TLR2 or TLR4. We have done experiments using TRL2/4 KO BMDCs to confirm these interactions, but we cannot get clear data. However, as mentioned in Line 347-349, we know that Rv2005c-mediated DC activation and maturation affected through the MAPK signaling pathway. Therefore, we could not explain how Rv2005c activates DC at present. It seems to be involved in another surface molecules in DC activation by Rv2005c or one possibility may be due to direct pinocytosis of the protein. By our experiences, some Mtb proteins can activate the macrophages or dendritic cells via TLR2 or TLR4, but many proteins can activate these cells without involvement of TLR2 or TLR4. Unfortunately, we have been focused on TLR2 and TLR4 molecules as an interacting partner of the mycobacterial proteins, we did not test other surface molecules interacting with Rv2005c. Because of your comments, we have realized that further detail investigation about mechanism or other molecules involved in DC or macrophage activation induced by the protein is required. We are going to study to prove a clear mechanism related with protein-mediated DC activation.

For issue of the protein purity, although it does not coincide in the condition reviewer suggested, we will show the data that confirmed the purity using ImageJ. As shown in Reference Figure 1, we were able to confirm that 92% protein was obtained. Of course not 100%, but we thought that this is the high purity% that can be obtained in the laboratory. Next, as your suggestion, the purified Rv2005c we have used in our study was subjected to MS analysis, but the amount of the protein sample left was too small to get reliable results.

Reference Figure 1. Confirmation of Rv2005c purity using ImageJ.

[Q2] I appreciate the corrections that in response to my question 2. However, I don’t see a change in the experiment starting in line 379 indicating that naïve CD4 T cells were primed and produced IFNg after co-culture with Rv2005-stimulated DCs. This is, as I wrote in my comment before, immunological not possible. (The T cell repertoire within naïve T cells is very broad to be able to recognize more or less any possible encountered antigen (estimations of around 10^6 to 10^8). To use naïve T cells to detect priming to one specific peptide or antigen in a cell culture assay is therefore to my understanding not possible (you may have none or max very few cells recognizing the presented peptide per well). This is why these assays are performed with mice from transgenic mice expressing one T cell receptor (=all responding to the presented antigen) as the authors did in in figure 2A. An alternative is to use T cells from infected mice if one is interested in recall responses to an expanded T cell population (this antigen must be triggering expansion) to have substantial T cell numbers to detect responses. Therefore, all experiments claiming naïve T cell priming after DC co-culture and also the co-culture with infected macrophages can not be connected to specific T cell activation but may be due to bystander activation or other mechanisms. This needs to be discussed or experimental set up should be re-thought (my recommendation). T cells from transgenic mice expressing specific T cell receptors recognizing presented Rv2005c peptides would be required to perform the experiments and draw the conclusions the authors did.)

[A2] First of all, the parts you indicated were corrected; the Naïve T cell in Line 379 was replaced with OT-II TCR transgenic, and the word “prime” was replaced with “coculture”. The expression of “Naïve” in Figure 2 C-D was also deleted and replaced with “pharmacological activated T cells”.

We partially agreed with your indication about naive T cell priming after DC co-culture. But we could not fully explain our results by only bystander activation or other mechanisms. Because Rv2005c expression in Mtb within macrophages was increased under hypoxic state, which induce intracellular Mtb growth inhibition by T cells activated by Rv2005c-matured DCs, but these activated T cells did not inhibit Mtb growth in macrophages under normxia. These results suggest that peptides derived from Rv2005c are specifically involved in T cells priming by Rv2005c-matured DCs or macrophage activation by these activated T cells. As your indicated, the bystander T cells may be nonspecifically involved, which is not enough to explain our results. And also previously published paper has reported that mycobacterial proteins-matured DCs could activate naïve T cell in vitro coculture system (PMID: 28193909, 20176745). Ratio of DC:T cells usually is 1:10 in coculture system and a protein-matured DCs can activate several naïve T cell clones because it has the multiple T cells epitopes, and then primed T cells further expand during another culture period.     

Naïve T cells only are used in priming experiment with Ag-matured DCs because some microbial agents cannot be challenged in human. So many “Naïve T cell priming in vitro” experiments are conducted (PMID: 19616202, 16670315). And it is reported that methods to prime human Naïve CD4+ T cells in vitro would be of significant value for the pre-clinical evaluation of vaccine candidates and other immunotherapeutics (PMID: 19925804).

Because of your comments, we have further confirmed whether naïve T cells primed by Rv2005c-matured DCs could really be primed. T cells from splenocytes cocultured with LPS- or Rv2005c-matured DCs were purified by MACS column and then cocultured with newly LPS-, Ag85- and Rv2005c-matured DCs. T cells primed with Rv2005c-matured DCs were specifically activated by newly Rv2005c-matured DCs, but not T cells primed with LPS-matured DC, indicating that LPS-matured DC could not activate any T cells. Ag85-matured DCs was used to determine antigen specific priming (Reference Fig. 2). We believe it would be possible to mimic the cellular events necessary to generate naïve T cell responses in the laboratory.

Reference Figure 2. Production of IFN-g by Rv2005c antigen-specific Naïve T cells activity upon stimulation with matured DCs expressing Rv2005c using ELISA. To examine Rv2005c-specific T cells activity, LPS-, or Rv2005c-matured DCs were co-cultured with splenocytes of naïve mice, at a DC to T cell ratio of 1:10. Then, T cells were isolated using a MACS column and newly immature DCs, LPS-matured DCs, Ag85B-matured DCs or Rv2005c-matured DCs were co-cultured with T cell primed with LPS-, or Rv2005c-matured DCs. The quantities of IFN-γ in the culture supernatant were determined by ELISA. The data shown are the mean values ± SD (n = 3); ∗∗∗p < 0.001 compared to each group.

Thank you again for your good comments and deep interest.
